# Code-Switching Metrics Using Intonation Units

**Rebecca Pattichis**[1]  **Dora LaCasse**[2]  **Sonya Trawick**[3]  **Rena Torres Cacoullos**[3]

[1] University of California, Los Angeles
[2] University of Montana
[3] Pennsylvania State University

`pattichi@cs.ucla.edu, dora.lacasse@mso.umt.edu,`
`sonyatrawickpsu@gmail.com, rena@psu.edu`

## Abstract

Code-switching (CS) metrics in NLP that are based on word-level units are misaligned with true bilingual CS behavior. Crucially, CS is not equally likely between any two words, but follows syntactic and prosodic rules. We adapt two metrics, multilinguality and CS probability, and apply them to transcribed bilingual speech, for the first time putting forward Intonation Units (IUs) – prosodic speech segments – as basic tokens for NLP tasks. In addition, we calculate these two metrics separately for distinct mixing types: alternating-language multi-word strings and single-word incorporations from one language into another. Results indicate that individual differences according to the two CS metrics are independent. However, there is a shared tendency among bilinguals for multi-word CS to occur across, rather than within, IU boundaries. That is, bilinguals tend to prosodically separate their two languages. This constraint is blurred when metric calculations do not distinguish multi-word and single-word items. These results call for a reconsideration of units of analysis in future development of CS datasets for NLP tasks.

## 1 Introduction

The mismatch between actual bilingual code-switching behavior and renditions of code-switching (CS) has been exposed in previous work. In particular, unnatural switching is characterized by the shortness of language spans and the arbitrariness of language boundaries (Bullock et al., 2019). Yet to date, metrics of CS complexity have been word-level based (e.g., a 4-word utterance with 3 switches, $w_{L1}$ $w_{L2}$ $w_{L1}$ $w_{L2}$, Gambäck and Das, 2016:1851). Such word-level metrics entail that any word is an equally likely site for language switching. In addition, they treat other-language single-word incorporations and alternating-language multi-word strings identically. Neither is a foolproof assumption. In this paper, we adapt Natural Language Processing (NLP) word-based metrics to account for prosodic-syntactic constraints on CS by using Intonation Units as tokens. We calculate two metrics: overall language distribution (multilinguality) and observed CS rate (probability). Results indicate that distinguishing single- and multi-word items in Intonation Unit-based metrics better captures CS constraints than amalgamating the two mixing types.

As a working definition, bilingual CS is going back and forth between two languages within a speaker turn, even within a prosodically and grammatically complete sentence, as in the following example, reproduced verbatim from the New Mexico Spanish-English Bilingual (NMSEB) corpus (Torres Cacoullos and Travis, 2018: Chs 2 & 3)[1]. See Transcription conventions in Fig. 1. Italic and roman type represent speech originally in English (E) and Spanish (S), respectively.

| | |
|---|---|
| ahí estaba con mi e- -- | S |
| con mi espejito, | S |
| que así me miraba y, | S |
| *if I could see my profile,* | E |
| ... y luego me volteé pa' atrás, | S |
| *I could see the back of my head.* | E |
| | |
| there I was with my m- -- | S |
| with my little mirror, | S |
| that I was looking at myself and, | S |
| *if I could see my profile,* | E |
| ... and then I turned around, | S |
| *I could see the back of my head.* | E |
| (03, 05:45-05:53) | |

---

[1]Within parantheses following each example are the NMSEB corpus transcript number and beginning-end time stamp. https://nmcode-switching.la.psu.edu

## 1.1 The Syntactic and Prosodic Structure of CS

Language is not composed of arbitrary strings of words, but is structured syntactically and prosodically. Syntactically, sequential word groups make up constructions and constituents (e.g., Bybee, 2010:136). For example, in the active and passive sentence versions, *they allow the new school to wear hats in the halls* and *the new school's allowed to wear hats in the halls*, "the new school" (article + adjective + noun) is a unit that as such is the object in one and the subject in the other (cf. Weiner and Labov, 1983:34).

Syntactic word groups are relevant for CS. According to the Equivalence constraint, CS is avoided at points of word order incompatibility (Poplack, 2013:586; Sankoff, 1998) (cf. Deuchar, 2020:255; Muysken, 2000:27, Pfaff, 1979:291). For example, in the Spanish-English language pair, articles are placed before nouns in both languages. However, there is a word order conflict for attributive adjectives, which are positioned before the noun overwhelmingly in English but only sometimes so in Spanish. As predicted by the Equivalence constraint, CS is far more likely after the article than between the adjective and noun (cf. Parafita Couto and Gullberg, 2019:702). Instances like "permiten el *new school to wear*" ('they allow the new school to wear') are quantitatively preferred (at a 4:1 ratio) over instances like "permiten el nuevo *school to wear*" (Torres Cacoullos and Vélez Avilés, 2023:17-18).

Prosodically, spoken language is broken into Intonation Units (IUs), which are speech segments "uttered under a single, coherent intonation contour" (Du Bois et al., 1993:47; cf. Chafe, 1994: 53-70; see Figure 2 below). In the NMSEB corpus, each line of transcription represents an IU, with punctuation marking the transitional continuity between IUs, or the terminal pitch contour. IUs are bound to the syntactic organization and processing of language. Words in the same IU tend to have a closer syntactic relationship than those in neighboring IUs (Croft, 1995:849-864), and IU boundaries are used in planning utterances (Ono and Thompson, 1995) and "tracking speech" (Inbar et al., 2020).

IUs are also relevant for CS. Corresponding to the Equivalence constraint—which requires local equivalence of word order in the two languages around a switch point—is the IU-Boundary

| . | final intonation contour |
|---|---|
| , | continuing intonation contour |
| -- | truncated intonation contour |
| - | truncated word |
| .. | short pause (0.2 secs) |
| ... | medium pause (0.3-0.6 secs) |

Figure 1: Transcription conventions (Du Bois et al., 1993). Each line represents an Intonation Unit (IU). For the purposes of readability, we have removed vocal noises, laughter and vowel lengthening, and excerpt-initial pauses. A prosodic sentence is an IU or series of IUs, containing at least one finite verb, that ends in intonational completion (Chafe, 1994:139).

constraint. This captures bilinguals' tendency to prosodically separate the two languages. The IU-Boundary constraint states that CS is favored across IU boundaries (cf. Torres Cacoullos and Travis, 2018: 51). That is, switching is far more likely between two words at the boundary of IUs, or across IUs, than between two words within the same IU. The prosodic and syntactic constraints are related in that the looser syntactic relationship between words at IU boundaries than within them may mean greater word order flexibility. However, the two constraints are independent in that, both at IU boundaries and within IUs, CS is subject to the Equivalence constraint and, among Equivalence points, CS is more likely at IU boundaries than within them.

Responding to these facts, we apply CS metrics using IUs rather than individual words as the basis.

## 1.2 Lone Items vs. Multi-Word Strings

Mixing is of two broad types: single-word incorporations, or lone items, as in the following example, and multi-word strings, or multi-word CS (as in the first example above). These types are approximately parallel to the distinction between CS of the insertional and alternational kind (Muysken, 2000; Muysken, 2015: 251–254), though they are operationally defined as one word and two or more words, respectively (on quantitative grammatical patterns of lone items, see Poplack and Dion, 2012; Sankoff et al., 1990).

tenían unos *desks* muy grandes,    SLS

they had some really big *desks*,    SLS
(03, 53:25-53:27)

The two types have divergent structural properties (Poplack, 2018; Torres Cacoullos and Travis, 2020:256-259). Lone items are disproportionately nouns, are placed according to the word order of the surrounding (recipient or matrix) language, and participate in the categories and constructions of that language (for example, lone English nouns like *desks* are given a Spanish gender category and may have a post-nominal adjective). In contrast, multi-word strings are placed at cross-language equivalence points (where the word order is the same) while the internal constitution of each string is consistent with the grammar of its respective language.

For many Spanish English bilingual communities, the single- vs. multi-word distinction is additionally important because the distribution of the two mixing types by language is starkly different. Language mixing with lone items is asymmetrical (Poplack, 2018). Lone items are 52% English nouns (le puso un *roof* nuevo 'she gave it a new *roof*' [25, 26:31]), but only 7% Spanish nouns (*put the whole* calabaza *there 'put the whole* pumpkin *there'* [17, 30:01]) (plus English and Spanish other than noun words, 21% apiece; NMSEB, $n$=2,991) (Torres Cacoullos and Vélez Avilés, 2023).

In contrast, multi-word CS is bidirectional (Torres Cacoullos and Travis, 2018:67-72). Constituting the 15% ($n$=2489) of eligible sentences (at least four words) hosting multi-word CS (alternating-language strings of at least two words) are 6% Spanish to English and 4% English to Spanish, with another 5% hosting more than one instance of CS. Examples are: pues estaban asina *como caddy corner across the street* 'well they were like [...]' [04, 1:02:12]); ~*Alma was more* flaca que la mama, '[...] thin than mama' [04, 1:04:18]); *bring – that name*, porque no había ~Roberts, todavía, in the family, '[...] *because there weren't any* ~Roberts, yet, [...]' [04, 15:59]).

Finally, lone other-language items and alternating multi-word CS are only moderately correlated. For the NMSEB corpus, a Pearson correlation coefficient assessing the relationship between the rate of lone items (14%, 2378/16957, of sentences host at least one lone item) and the rate of multi-word CS (15%, 2489/16957, of sentences host at least one multi-word CS) yields a positive correlation of just $r(38) = .58, p < .01$. For this set of reasons, we will distinguish lone items and multi-word strings in applying the CS metrics.

## 2 Related Methods

Recent work in NLP is concerned with incorporating, understanding, and generating CS text. Although there are already large multilingual language models such as M-BERT (Pires et al., 2019) and XLM-RoBERTa (Conneau et al., 2019), these LLMs were nevertheless trained on parallel monolingual data, which does not accurately capture the nuance of likely switch points in code-switched language inputs. CS datasets, on the other hand, are scarce and noisy, putting CS modelling in the domain of low-resource tasks (Doğruöz et al., 2021). Recently, the LinCE benchmark compiled by Aguilar et al. (2020) provides different CS datasets for Spanish-English, Hindi-English, and other language pairs for the tasks of Language Identification, Part of Speech Tagging, Named Entity Recognition, and Sentiment Analysis. Often, datasets are collected and scraped from social media (i.e., Twitter). However, there are two problems with this collection approach.

First, CS is known to happen in community settings among interlocutors with similar CS patterns (cf. Deuchar, 2020; Poplack, 2018). Therefore, mass collecting short tweets does not necessarily provide a true representation of intra-community CS. Second, these datasets do not hold a distinction between mixing types. Specifically, without this distinction, examples that merely use lone items tend to represent a majority of CS examples. In other words, most CS datasets used in NLP inherently rely on asymmetric models such as the Matrix Language Frame (MLF) model, which assumes a base language for the grammatical frame with words or phrases embedded from the other language (Myers-Scotton, 2002). Instead, the Equivalence constraint and IU-Boundary constraint accommodate alternating-language multi-word strings. Here, we showcase the different perspectives that the treatment of lone items provides on CS.

Barnett et al. (2000) developed the Multilingual-Index (M-Index) as a measure of the multilinguality of different corpora, or the distribution of languages in a corpus. Guzman et al. (2017) also created the Integration-Index (I-Index), which is meant to measure the probability of CS in different multilingual corpora. Mave et al. (2018) use both of these metrics in the creation and validation of their code-switched Twitter dataset for Hindi-English and Spanish-English pairs. However, it is

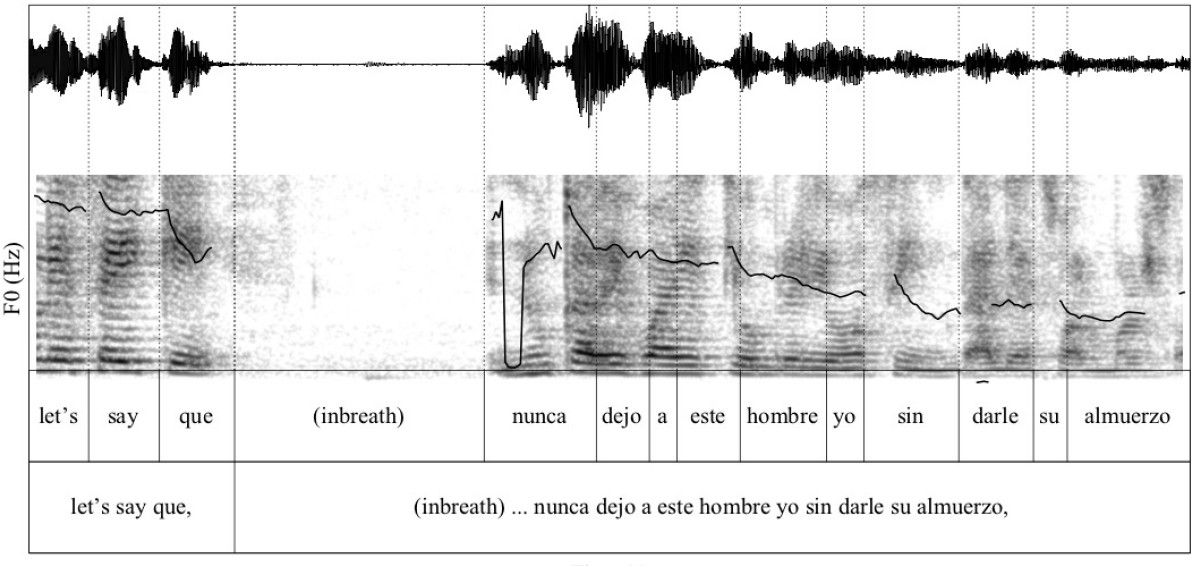

| let's | say | que | (inbreath) | nunca | dejo | a | este | hombre | yo | sin | darle | su | almuerzo |

| let's say que, | (inbreath) ... nunca dejo a este hombre yo sin darle su almuerzo, |

Time (s)

Figure 2: Acoustic properties of Intonation Unit include higher pitch at the beginning of the IU and slower rate of speech at the end of the IU, and sometimes pausing betweens IUs.

important to note that the I-Index was originally developed with the purpose of analyzing continuous text. Therefore, disjointed and small input texts such as tweets might not be a good use case.

## 3   Methods

We use both M-Index and I-Index in our experiments with continuous and majority monological transcripts of intra-community speech. To our knowledge, this is the first study to use the Intonation Unit (IU) as the token of measurement for NLP analysis of CS datasets. With this work, we highlight the different perspectives of CS – separating lone items from multi-word strings – and present the IU as a relevant token for future NLP dataset construction.

### 3.1   Description and Selection of Corpora

The NMSEB corpus (Torres Cacoullos and Travis, 2018: Chapters 2 and 3) provides the data for this project. The speakers are bilingual members of a long-standing speech community in northern New Mexico where Spanish and English have both been spoken for 150 years. Though New Mexican Spanish is endangered by shift to English as well as disparagement in comparison with textbook Spanish, these speakers regularly use both languages in their daily interactions. This balance of languages is reflected in the even distribution of prosodic sentences by language (43% English, 42% Spanish, 15% both, $n$ = 16,957). The integrity of both lan-

guages is seen in quantitative patterns of grammatical structures in each language, which align with their respective monolingual benchmarks (Torres Cacoullos and Travis, 2018: 204).

The recordings are transcribed orthographically and prosodically in IUs (Du Bois et al., 1993:47). IUs are reliably identified by their acoustic correlates, which include a rising or high starting pitch that falls as the IU progresses, faster speech at the start of the IU as compared to the end, and, often, pauses between IUs (Figure 2).

Directly bearing on the problem at hand, CS is structured on the basis of prosodic boundaries and terminal pitch contours (Torres Cacoullos and Travis, 2018:51-52 and references therein, among others). The first example in the following pair illustrates multi-word CS at the boundary of IUs (across-IU CS); the second one illustrates multi-word CS within a single IU (within-IU CS). As seen in Figure 3, CS is four times more likely at the boundary of IUs than within them. Furthermore, CS at an IU boundary is more likely following final than continuing intonation (cf. the distinction between inter-sentential and intra-sentential CS). This prosodic structure of multi-word CS is represented in the IU-Boundary constraint: prefer CS across IU boundaries. That is, bilinguals tend to keep their two languages prosodically separate.

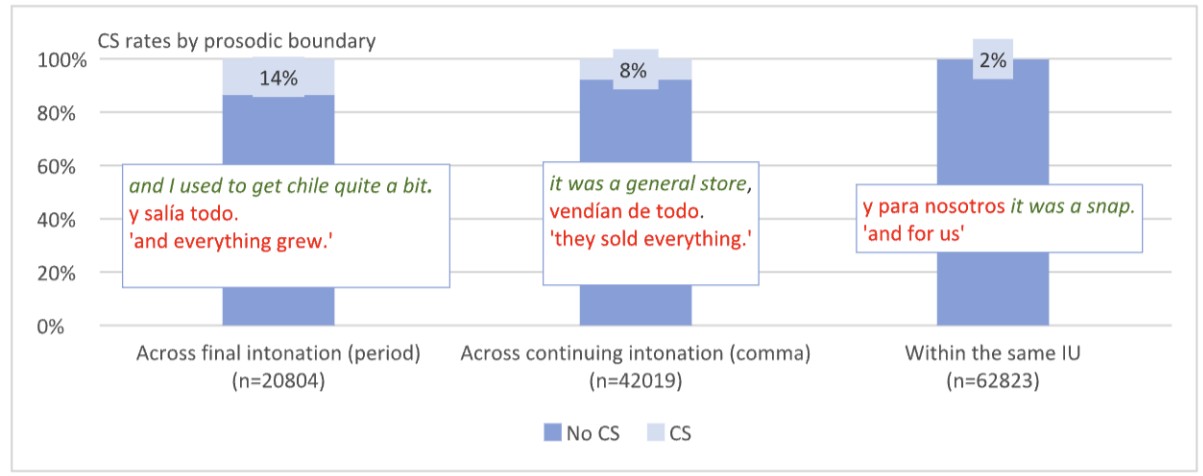

Figure 3: IU-Boundary constraint: Bilinguals quantitatively prefer multi-word CS across Intonation Unit (IU) boundaries (adapted from Trawick, 2022: §3.4).

| | |
|---|---|
| *it was a general store,* | E |
| vendían de todo. | S |
| | |
| *it was a general store,* | E |
| they sold everything | S |
| (03, 30:10-30:12) | |

---

| | |
|---|---|
| llegaba mi tío *the following days,* | SE |
| | |
| my uncle arrived *the following days,* | SE |
| (03, 40:51-40:53) | |

The present study draws on five transcribed recordings, totaling 4.8 hours and around 41,000 words, or 14,000 IUs. They are largely monologic, which allows us to treat CS as occurring within the same speaker turn (not in response to "interactive alignment" with an interlocutor, see, e.g., Kootstra et al., 2020). The speakers are: transcript 03, Sandra (administrator, Española); 05, Rocío (teacher aid, Santa Fe); 10, Pedro (school administrator, Taos); 16, Manuel (electrician and rancher, Rio Arriba); and 27, Eduardo (general contractor and store owner, Rio Arriba).[2] In the transcripts chosen, 84% - 97% of the IUs are produced by the speaker rather than the interviewer or another interlocutor, satisfying our threshold for majority monological

---

[2]Names given are pseudonyms and locations listed are either counties or major cities to protect speaker privacy. Anonymization also occurred within each transcript, so that any real names, nicknames, or identifiable proper nouns (i.e., small cities, places of work, high school names, etc.) were replaced and indicated with a preceding "~" (Torres Cacoullos and Travis, 2018:48).

speech.

We count all IUs, with the exception of IUs consisting of language-neutral material (fillers such as *uh*, backchannels such as *yeah*) (Torres Cacoullos and Travis, 2018:51). IU rows were considered as eligible for future analysis if they contained a language tag of either 'S' (Spanish), 'E' (English), or 'L' (Lone Item). Note that a language tag may be a combination of 'S', 'E', or 'L'; see previous examples for illustrations of language tagging.

### 3.2 Quantitative Experiments on the IU-Token Level

In order to lay bare how amalgamating lone items with multi-word strings affects perspectives on CS, we compute metrics (i.e., M-Index (Barnett et al., 2000) and I-Index (Guzmán et al., 2017)) for different representations of the corpora (see Tables 1-3). Specifically, I-Index measures were calculated two ways: by considering only 'S' or 'E' for analysis, and by also including 'L' (re-coded as 'S' or 'E').

Since we are operating at the IU-token level, each token can contain more than one language tag if there is a within-IU language switch. Below we describe the binary measures that allow us to maintain the integrity of our token level.[3]

#### 3.2.1 M-Index

The Multilingual Index (M-Index) proposed by Barnett et al. (2000) is meant to measure the multilinguality of a given corpus with at least two languages from a range of 0 to 1, where the former is

---

[3]Code can be found at https://github.com/rpattichis/IU-Boundary_constraint_code.

monolingual, and the latter means there is a perfect balance of languages. Here, $k$ denotes the number of languages in the corpus, and $p_j$ is the number of tokens in language $j$ divided by the total number of tokens:

$$\text{M-Index} = \frac{1 - \sum p_j^2}{(k-1) \cdot \sum p_j^2}.$$

While it is originally meant for the word-token level, we use it at the IU level. That is, instead of the numerator of $p_j$ representing the number of words in language $j$, we instead make $p_j$ the amount of IUs in language $j$ divided by total IUs. When an IU has multiple languages contained within its bound, we tag it with the earliest language present (i.e., an IU with 'SES' will count as 'S'). For the M-Index, we only considered 'S' and 'E' as valid language tags.

### 3.2.2 Across-IU I-Index

Here, we use the Integration Index (I-Index) developed by Guzman et al. (2017) to measure the probability of CS in each transcript. Specifically, the I-Index is meant to approximate the probability that any given token is a switch point. Here, $n$ is the number of tokens, and $S(l_i, l_j)$ is 1 if two neighboring tokens are in different languages, 0 otherwise:

$$\text{I-Index} = \frac{1}{n-1} \sum_{1 \leq i = j-1 \leq n-1} S(l_i, l_j).$$

Again, while this metric was developed with an assumption of words as tokens, we count the IUs as tokens. Then, for the across-IU I-Index, we ask the question: Does a language switch happen at the boundary between the $i^{th}$ IU and the $(i+1)^{th}$ IU? This binary measure determines the value of $S(l_i, l_j)$. Here, we consider two perspectives: first, with only the 'S' and 'E' language tags, and then with the inclusion of 'L' to understand how lone items impact switching across IUs.

### 3.2.3 Within-IU I-Index

To also account for the within-IU CS that occurs in our corpora, we use the same I-Index but change our question: Does a language switch occur within the $i^{th}$ IU? Although there might rarely be more than one switch point within an IU, we decided to keep the binary measure so as to not double count a token. Here, the only change is that $S(l_i)$

only looks at one token, rather than a comparison between two tokens. Again, for this metric, we consider the two perspectives with only 'S' and 'E' as well as the inclusion of 'L' to understand how including lone items may impact our understanding of CS.

Note that although later work by Bullock et al. (2019) propose a normalized I-Index due to its dependence on a corpus's M-Index, we have intentionally chosen transcripts that are comparable in their M-Index. Specifically, the M-Index is close to 1 for three transcripts (03, 10, 16) and .50 for two (05, 27).

## 4 Results

Table 1 gives the number of IUs counted as 'S' and as 'E' and the number of IUs hosting CS according to prosodic position and the treatment of lone items.[4] Table 2 gives the corresponding percentages. M-Index and I-Index for the five transcripts appear in Table 3. Of the five corpora chosen, three have an M-Index close to 1, indicating a balance of English and Spanish within the transcript. In particular, M-Indices of .94-.97 correspond to speakers who produce a more balanced 41%-45% of their IUs as 'S' and 55%-59% as 'E', while values of .52-.56 resulted for speakers with 22%-23% in one language (and thus more than 75% in the other).

As shown in Table 3, the combination of the M-Index and I-Index is crucial in understanding the nuance of different speakers' CS patterns. For example, transcripts 03 and 16 have similar M-Indexes (with a difference of 0.03), but the former has almost twice the I-Index of the latter speaker. This indicates the independence of the two metrics.

We also use language distribution graphs to visualize the M-Index and I-Index within a transcript, in Figure 4. Here, in an ordered array of the language tag for each IU token, English IUs are colored in purple, whereas Spanish is in yellow. These visualizations help elucidate what the quantitative metrics distinguish, by the number and width of the language bands. For example, transcripts 03 and 10 (language distribution graphs in Fig. 4c and 4e, respectively), have a similar M-Index and Across-IU I-Index, which is evident from their language distribution graphs. On the other hand, 05 (Fig. 4a) has a patently different M and I-Index from these

---

[4]Counts in 'IUs w/ Ls' column do not correspond to the differences between 'Ls incl.' and 'no Ls' in the preceding columns because an 'L' at an IU boundary may count as both an Across- and Within-IU switch.

| Corp | Total S/E | Across IU | | W/in IU | | IUs |
|---|---|---|---|---|---|---|
| | | no Ls | Ls incl. | no Ls | Ls incl. | w/ Ls |
| 05 | 1911/532 | 77 | 101 | 5 | 23 | 23 |
| 27 | 616/2035 | 194 | 238 | 12 | 58 | 54 |
| 03 | 1040/1501 | 376 | 464 | 29 | 112 | 111 |
| 16 | 994/1266 | 189 | 252 | 17 | 71 | 70 |
| 10 | 737/894 | 264 | 276 | 21 | 49 | 31 |

Table 1: Number of IUs counted as 'S' and 'E'; number of IUs hosting CS by prosodic position—Across IU boundaries vs. Within IU—and by treatment of lone items—'No Ls' vs. 'Ls incl.'; IUs hosting Ls.

| Corp | % S/E | Across IU | | W/in IU | | IUs |
|---|---|---|---|---|---|---|
| | | no Ls | Ls incl. | no Ls | Ls incl. | w/ Ls |
| 05 | 78/22 | 3.2 | 4.1 | 0.2 | 0.9 | 0.9 |
| 27 | 23/77 | 7.4 | 8.9 | 0.5 | 2.2 | 2.0 |
| 03 | 41/59 | 14.8 | 18.0 | 1.1 | 4.3 | 4.3 |
| 16 | 44/56 | 8.4 | 11.0 | 0.8 | 3.1 | 3.1 |
| 10 | 45/55 | 16.2 | 16.8 | 1.3 | 3.0 | 1.9 |

Table 2: Percentages corresponding to Table 1.

| Corp | M-Index | I: Across | | I:W/in | |
|---|---|---|---|---|---|
| | (S/E) | no Ls | Ls incl. | no Ls | Ls incl. |
| 05 | 0.52 | 0.03 | 0.04 | 0.0 | 0.01 |
| 27 | 0.56 | 0.07 | 0.09 | 0.0 | 0.02 |
| 03 | 0.94 | 0.15 | 0.18 | 0.01 | 0.04 |
| 16 | 0.97 | 0.08 | 0.11 | 0.01 | 0.03 |
| 10 | 0.98 | 0.16 | 0.17 | 0.01 | 0.03 |

Table 3: M-Index and I-Index; I-index calculated for CS according to prosodic position and according to inclusion of Lone items (see methods section).

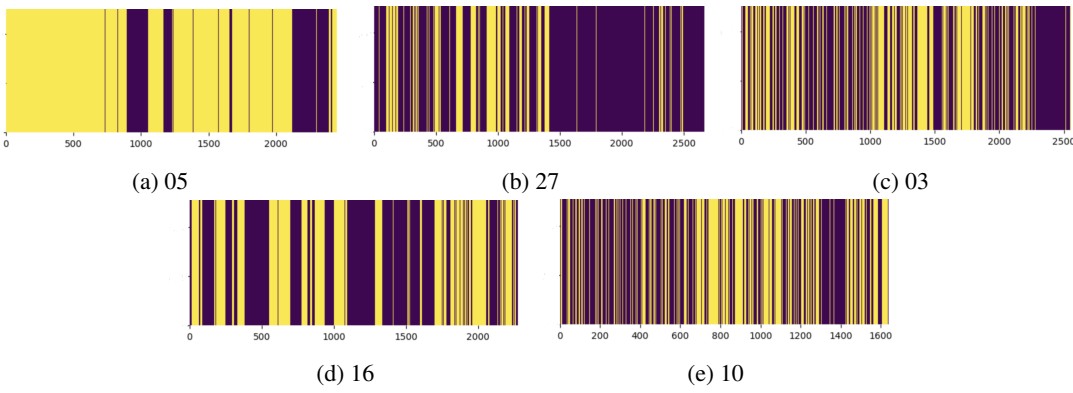

(a) 05      (b) 27      (c) 03

(d) 16      (e) 10

Figure 4: Language distribution graphs for each transcript, where English is in purple and Spanish is in yellow.

transcripts, resulting in a smaller total number of language bands and wider bands for Spanish.

As to how the inclusion of lone items impacts our perspective of CS, it seems to have little impact on the Across-IU I-Index, as seen in Table 3. However, including the 'L' language tag does have a substantial impact on the Within-IU I-Index.

While for the Across-IU index the increase is at most 37.5% (for 16, from 0.08 to 0.11), for the Within-IU index the increase is as large as 300% (for 03, from 0.01 to 0.04) (disregarding the cases of infinity, for 05 and 27). This makes sense – lone items occur within a single IU (and are usually nouns positioned at the end or interior of the IU

with fewer than 15% occurring IU-initially (Steuck, 2018:87)).

Furthermore, merging the single-word items with multi-word CS not only substantially raises the within-IU I-Index, but also magnifies differences between speakers with respect to the within-IU I-Index. Within-IU I-Indices range from 0 to 0.01 with 'no Ls,' but from 0.01 to 0.04 when lone items are included. This is important because it shows that with the latter perspective (i.e., merging of lone items with multi-word CS), we fail to capture the IU-Boundary constraint on CS whereby speakers strongly prefer switching across prosodic units rather than within them.

## 5 Conclusion

CS metrics are appropriate when they are grounded in spontaneous speech within bilingual communities, the natural habitat of CS. We have illustrated how using the prosodic structure of CS and distinguishing different mixing types achieves this.

We put forward the Intonation Unit as the token level for CS metrics. In addition, we distinguish multi-word CS from single-word (lone) other-language items. Application of the IU-based metrics captures the tendency for multi-word CS to be used from one IU to another in across-IU CS. All speakers (regardless of their M-Index or Across-IU I-Index) strongly disfavor within-IU multi-word CS. This is the IU-Boundary constraint of bilingual speech. In contrast, lone items, which by definition are used within a single IU, notably raise the I-Index for within-IU CS. Merging multi-word CS and lone items thus obscures uniform adherence to the IU-Boundary constraint, despite individual differences in language distributions (M-Index) and CS rate (I-Index).

With these findings, we highlight the IU as an important token level for future CS datasets, as well as the impact of single-word (lone) items as distinct from multi-word strings on CS metrics. We see relevance for this work in Controlled Natural Language Generation (NLG). Much of the CS work is slow/iterative because it is hard to find quality CS data (Tarunesh et al., 2021; Doğruöz et al., 2021). Thus, our work impacts the future construction of transcribed CS datasets, specifically with the injection of linguistically well-founded CS patterns to improve current NLG methods. Currently, CS generation has focused on word substitutions (Solorio and Liu, 2008; Tarunesh et al., 2021). We

have shown here that bilingual behavior is not limited to word substitution. In accounting for multi-word language alternations, future NLP metrics can draw on the Equivalence constraint (syntax) and the related but independent IU-Boundary constraint (prosody).

In sum, this paper is the first to use IUs for CS metrics. The results provide critical insight for future transcription and synthetic data generation methods that can improve CS datasets, which will ultimately impact all downstream NLP tasks.

## 6 Limitations

Multi-word tokens (whether IUs, clauses or some other well-defined multi-word unit) yield CS metrics that are more in line with actual CS patterns. Nevertheless, due to differences in how tokens are delimited (word-based vs. larger units), direct comparisons between results in this and previous word-based studies remain to be shown. Moreover, as noted in prior work, CS metrics are still fairly crude (e.g., it is not known what degree of difference in an M or I index would be significant). Since factors such as the length and syntax of a multi-word token likely affect the probability of CS, ways need to be found to incorporate predictors and constraints on CS into more refined metrics.

## 7 Ethical Considerations

To protect privacy, speaker names given are pseudonyms and locations listed are major cities or counties (not specific towns). Within each transcript, any real names, nicknames, or identifiable proper nouns (small cities, places of work, high school names, etc.) are also anonymized (Torres Cacoullos and Travis, 2018:48). We give speaker information, even if it does not seem relevant to the results, as a reminder that the data are instances of real bilingual speech (i.e., not synthetically generated), grounded in a well-defined community in New Mexico. These data are not confined by the scope of accessibility issues (i.e., technology access, as with Twitter data), and therefore represent a diverse population not limited to urban, well-educated, upper-class professionals or elite bilinguals.

## Acknowledgements

Support from the National Science Foundation (BCS-1019112/1019122, 1624966) is gratefully acknowledged.

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
