# OpenReview forum: "Code-Switching Metrics Using Intonation Units"
_EMNLP/2023/Conference — EMNLP 2023 Main_

### Official Review · Reviewer_AZWa · 2023-08-03

**Soundness:** 3

**Ethical Concerns:**

Yes

**Excitement:**

3: Ambivalent: It has merits (e.g., it reports state-of-the-art results, the idea is nice), but there are key weaknesses (e.g., it describes incremental work), and it can significantly benefit from another round of revision. However, I won't object to accepting it if my co-reviewers champion it.

**Justification For Ethical Concerns:**

Lines 297-301: These are 03 (Sandra, administrator, Española), 05 (Rocío, teacher aid, Santa Fe), 10 (Pedro, school administrator, Taos), 16 (Manuel, electrician and rancher, Rio Arriba), and 27 (Eduardo, general contractor and store owner, Rio Arriba)"
If these are real names and professions, the identity of these people can potentially be disclosed.

**Paper Topic And Main Contributions:**

This paper analyzes the role of IU Spanish and English CS using two metrics. One of them is M-Index, which measures how multilingual a corpus is. The other one is I-Index, which measures the probability of CS in each transcript. I-Indexes were further divided into two: within-IU index, and across-IU index. Results show that CS tend to occur not within but across IU boundaries.

**Questions For The Authors:**

Why was the information of Figure 1 put that way, and not as a table?  Moreover, it seems to me that only "," and "." are relevant.
Lines 154-155: "Lone items tend to be English incorporations into Spanish." Line 162 "In contrast, multi-word CS is bidirectional." Why are these discrepancies observed?
From line 176, a Pearson correlation is mentioned. How was it calculated (i.e., a function in Python)?
Most of the related methods section argues that Twitter data is not adequate for CS analysis. Why was such big emphasis put on Twitter?
Lines 453-455: "Metrics for predicting language distributions and CS probability are most appropriate when they cor- respond to bilingual speech on the ground."
What does "on the ground" mean here?
Why do the tables focus only on the I-Index and not the M-Index?
Why were the heatmaps of only 3 speakers shown? It might be interesting to have the heatmaps of the other ones in the appendix.

**Reasons To Accept:**

This paper offers relevant contributions to several areas of linguistics, such as to the discussion on linguistic representations (in this case, the IU), and also to issues related to bilingualism (in this case, on quantitative analysis of bilingualism). The lit review is comprehensive and the paper is methodologically sound. The explanations on the metrics are also very clear.

**Reasons To Reject:**

The main result of this paper is that speakers are more likely to perform CS across IU and not within them. Nonetheless, the authors argue about the relevance of the IU. Could that not indicate that the IU is not a relevant unit, or that the boundaries of the corpus is not reliable? Lines 268-273 state: "The recordings are transcribed orthographically and prosodically in IUs (Du Bois et al., 1993:47). IUs are reliably identified by their acoustic correlates, which include a rising or high starting pitch that falls as the IU progresses, faster speech at the start of the IU as compared to the end, and, often, pauses between IUs". Was the IU transcription checked by the authors?
There are also some sentences that are too long and could be reworded (see "Typos Grammar Style And Presentation Improvements" for more details). Name and professions of some speakers were revealed (see "Justification For Ethical Concerns"). If these were their real names, they could be identified.

**Reproducibility:**

2: Would be hard pressed to reproduce the results. The contribution depends on data that are simply not available outside the author's institution or consortium; not enough details are provided.

**Reviewer Confidence:**

5: Positive that my evaluation is correct. I read the paper very carefully and I am very familiar with related work.

**Typos Grammar Style And Presentation Improvements:**

Lines 77-82: "For example, in the Spanish-English language pair, whereas articles are placed before nouns in both languages, there is a word order conflict for attributive adjectives, which are positioned before the noun overwhelmingly in English but only sometimes so in Spanish."
This sentence is too long.

87-88 (‘they allow the new school to wear’)
Sometimes the authors use single quotes and sometimes double quotes. It would be nice to have them unified.

110-115: "Thus, for CS, corresponding to the syntactic Equivalence Constraint, which requires local equiv- alence of word order in the two languages around a switch point, is the IU-boundary Constraint, which captures bilinguals’ tendency to prosodically sep- arate the two languages."
This sentence is too long.

194-204 "Recently, the LinCE benchmark compiled by Aguilar et al. 198 (2020) provides different CS datasets for Spanish- English, Hindi-English, and other language pairs 200 for the tasks of Language Identification (LID), Part of Speech Tagging (POS), Named Entity Recognition (NER), and Sentiment Analysis (SA)."
Maybe the abbreviations are not necessary, since they appear only once.

Line 226: miltilinguality should be multilinguality

In figure 2, there is a problem with the pitch contour of "nunca"

Line 257: the word denigrate can be considered to have a racist etymology, so I would recommend using another word, such a desuse.

The authors use capital letters after colons (such as in line 289). I would recommend using lower case.

I assume that the numbers after the sentences at the end of page 4 represent a speaker number and a time stamp. It would be nice to have it clear in the paper.

Line 356: th (and other occurrences of it) would better in superscript.

---

> ### Author Rebuttal · Authors · 2023-08-28
>
> We appreciate Reviewer AZWa’s thoughtful engagement with our work, and for agreeing with Reviewer VWCM that our literature review is thorough and our methodology sound. We would like to point out, however, that Reviewer AZWa’s reasons to reject are either based on misunderstanding our results, or grammatical preferences that we are confident we can modify for improved readability. Additionally, they raise a valid ethical concern about speaker confidentiality, which we will also respond to. Our rebuttal will be formatted in the order of addressing misunderstandings, ethical concerns and reproducibility, manuscript clarifications, and formatting suggestions.
>
> _“Could that not indicate that the IU is not a relevant unit, or that the boundaries of the corpus is not reliable?”_
> No. Recall that our findings validate that switching happens more often at the IU boundary versus within the IU (lines 464-469). This means, then, that having information about where the IU boundaries lie is critical for improving predictability of a switchpoint (instead of assuming that a switch could happen with the same probability between any two given words).
>
> _“Was the IU transcription checked by the authors?”_
> We have no reason to question the transcription of this corpus. The corpus compilers report that the transcriptions went through at least five rounds;  each hour of recorded speech took at least 50 hours for transcription  (Torres Cacoullos and Travis, 2018: 47).
>
> _“Name and professions of some speakers were revealed (see "Justification For Ethical Concerns"). If these were their real names, they could be identified.”_
> We will add a statement that all speakers' names are pseudonyms in this corpus. There was also further anonymization within each transcript of names and nicknames of people mentioned as well as of locations (e.g., smaller towns, places of work, high school names, etc.) that could be identifiers (Torres Cacoullos and Travis, 2018:48). In short, our main motivation in including the speakers’ pseudonyms and basic demographic information is to remind readers that these transcripts are from real people representative of a well-defined bilingual community, and thus not limited to urban, well-educated, upper-class professionals or elite bilinguals. For a full justification as to why we would like to keep speakers’ pseudonyms, please see our rebuttal to the meta-ethics reviewer.
>
>
> We disagree with the reproducibility score given by Reviewer AZWa. The corpus has not been published due to the original ethical commitment to the speakers, given the intimate nature of the recordings by in-group interviewers in these close-knit communities. In addition, adopting protocols for access protects communities with a highly-stigmatized minority language variety from unintentional misinterpretation of local vernaculars and from publication of stereotyping examples (Torres Cacoullos and Travis, 2018:48). Penalizing the paper because the corpus is not posted on the web only further incentivizes the mass collection of web data, which is not fully representative of bilingual communities (see paragraph on ethics justification above). We plan to make the code completely open source through Github upon acceptance into the conference.
>
>
> _"Lines 154-155: "Lone items tend to be English incorporations into Spanish." Line 162 "In contrast, multi-word CS is bidirectional." Why are these discrepancies observed?"_
> The point here is simply to illustrate how the two mixing types differ in language distributions (lines 150-153). The asymmetry of lone items, which here tend to be English incorporations into Spanish, has been reported as a community-wide preference, and has been observed in other communities (Poplack, 2017). The bidirectionality of CS may be interpreted as an indicator of bilingual ability (Torres Cacoullos & Travis, 2018:67-72; Poplack, 1980).
>
>
> _"From line 176, a Pearson correlation is mentioned. How was it calculated (i.e., a function in Python)?"_
> Calculation was with the Pearson Correlation Coefficient Calculator, on the online Social Science Statistics website.
>
>
> _"Most of the related methods section argues that Twitter data is not adequate for CS analysis. Why was such big emphasis put on Twitter?"_
> To our knowledge, the Linguistic Code-Switching Evaluation (LinCE) benchmark is currently the only benchmark for downstream CS tasks (Aguilar et al., 2020). For three of the four tasks (which we refer to in lines 197-203), the datasets come from CS examples found on Twitter.
> The attraction of Twitter is its accessibility, but we hope to have made clear the potential advances to be made by working with speech data, which is the natural habitat of CS. Note, too, that the metrics we adapt were originally implemented on continuous streams of text (i.e., oral transcripts and written published books) (Guzmán et al., 2017: 70), and are therefore not suited for application to discontinuous and smaller spans of text like Twitter.
>
>
> _"Lines 453-455: "Metrics for predicting language distributions and CS probability are most appropriate when they correspond to bilingual speech on the ground." What does "on the ground" mean here? "_
> Here, we mean grounded in the patterns of real bilingual speech, observed in everyday conversations within the community. We will be sure to clarify this point after line 455.
>
>
> _"Why was the information of Figure 1 put that way, and not as a table? Moreover, it seems to me that only "," and "." are relevant."_
> We can modify the presentation of the transcription rules to make it more readable as a table. We include the transcription rules for transitional continuity (“.” “,” “--” “?”) because these demarcate IUs, as seen in the examples throughout the paper. We include the transcription rules for pausing (...), because pausing is also a form of prosodic separation between languages.  We are open to adding the modified table to the appendix.
>
>
> _"Why do the tables focus only on the I-Index and not the M-Index?"_
> The M-Index is mentioned in the second column of Table 3. We will modify the formatting of this table to better distinguish between the two metrics.
>
>
> _"Why were the heatmaps of only 3 speakers shown? It might be interesting to have the heatmaps of the other ones in the appendix."_
> We ultimately chose these of the 5 due to space limitations as well as overall readability, but we can include the other 2 visualizations in the appendix.
>
>
> _“In figure 2, there is a problem with the pitch contour of "nunca"”_
> The pitch contour was produced in Praat. A dip as in “nunca” may be capturing nonlinguistic information, as when a speaker’s voice gave out if they were speaking quietly.
>
>
> _"Line 257: the word denigrate can be considered to have a racist etymology, so I would recommend using another word, such a desuse."_
> We will replace it with disparage.
>
>
> _“I assume that the numbers after the sentences at the end of page 4 represent a speaker number and a time stamp. It would be nice to have it clear in the paper.”_
> Yes, we will clarify this.
>
>
> Lastly, we will work to improve the readability of the sentences deemed too long. We want to thank Reviewer AZWa again for their thorough comments.

---

### Official Review · Reviewer_VWCM · 2023-08-04

**Typos Grammar Style And Presentation Improvements:** 12
**Soundness:** 3

**Excitement:**

3: Ambivalent: It has merits (e.g., it reports state-of-the-art results, the idea is nice), but there are key weaknesses (e.g., it describes incremental work), and it can significantly benefit from another round of revision. However, I won't object to accepting it if my co-reviewers champion it.

**Paper Topic And Main Contributions:**

The present paper analyzes code-switching patterns in the New Mexico Spanish-English Bilingual corpus by Torres Cacoullos and Travis (2018). The corpus is analyzed using two slightly adjusted code-switching metric that measure multilinguality (What is the balance between the languages used?) and code-switching probability (How often does code switching occur?).

The main contribution of the paper is a thorough discussion of the phenomenon of code switching and the observation that multi-word code switching typically occurs at the boundary between two prosodic units rather than within a single prosodic unit, i.e., the two languages are typically prosodically separated. On the other hand, the incorporation of single words (typically nouns) of one language into a sentence of the other language is shown to be subject to known syntactic but not prosodic constraints. Therefore, the paper argues for distinguishing between single-word incorporations and multi-word code switching, and and for using prosodic units as tokens for analyzing multilinguality and code-switching probability, rather than word tokens.

**Questions For The Authors:**

1) I do not have access to the book and corpus by Torres Cacoullos and Travis (2018). Could you try to summarize what exactly is new in your paper compared to the findings in the book? The original corpus is segmented into prosodic units and distinguishes between single-word and multi-word code-switching. Assuming that your finding about the relevance of prosodic units is new, what was the reason for annotating them in the original corpus?

2) You show that, although less common, code-switching can also occur within prosodic boundaries. Could it be that the preference for code-switching at prosodic boundaries is simply caused by the Equivalence Constraint (because word order compatibility is highest at the boundary between two prosodic units)?


**Reasons To Accept:**

The paper analyzes its topic in a thorough, solid, and linguistically well-founded way. It also provides a good literature overview of code switching. It addresses the widespread phenomenon of code-switching, which is rarely addressed or considered in the field of NLP, and lays a solid foundation for dealing with it in a theoretically informed way.

**Reasons To Reject:**

The paper is limited to corpus analysis and it does not show or discuss how to apply its findings to NLP.

The paper analyzes the corpus described in the monograph by Torres Cacoullos and Travis (2018). Unfortunately, I do not have access to the full book, but based on reviews, it appears that at least some of the observations made in this paper have already been described or at least anticipated in the original book. For example, the original corpus is segmented into prosodic/intonational units, and a distinction is apparently made between single-word and multi-word code-switching.

**Reproducibility:**

5: Could easily reproduce the results.

**Reviewer Confidence:**

4: Quite sure. I tried to check the important points carefully. It's unlikely, though conceivable, that I missed something that should affect my ratings.

---

> ### Author Rebuttal · Authors · 2023-08-27
>
> We want to thank Reviewer VWCM for their thorough review. We especially appreciate their opinion that the paper should be accepted for our in-depth discussion of code-switching (CS) and theoretically sound manner of addressing CS (which is often overlooked in NLP). We want to emphasize our novelty beyond a thorough discussion on CS, however. Prior work relating to CS in NLP (specifically in generation and predicting switchpoints) has focused on word-level tokens (Solorio & Liu, 2008; Chang et al., 2019; Tarunest et al., 2021). Importantly, this paper is the first to use Intonation Units (IUs) for CS metrics, opening up a new line of work beyond the word-level unit. We will be sure to emphasize this novelty in our abstract.
>
> _“The paper is limited to corpus analysis and it does not show or discuss how to apply its findings to NLP.”_
> Reviewer CGqL mentioned this as well, and we address this concern in our rebuttal to them. CS is above all a phenomenon of informal speech among bilingual community members. Thus, corpus analysis is a prerequisite for developing applications. However, corpus analysis within NLP has been largely absent, precisely due to “the scarcity of corpora containing large volumes of real code-switched text” (Tarunesh et al., 2021). Much of the CS work is slow/iterative because it is hard to find quality CS data (refer to lines 195-197, where we cited (Doğruöz et al., 2021)). Our results, then, provide critical insight for future transcription and synthetic data generation methods that can improve CS datasets (see the answer to #2 on Reviewer CGqL for more details). We will be sure to emphasize this point in our literature review and conclusion.
>
> Reviewer VWCM’s second reason for rejection calls into question the novelty of our work with respect to previously published work using the same corpus, especially
> Torres Cacoullos and Travis (2018). We will dive into these concerns below.
>
> - _“Could you try to summarize what exactly is new in your paper compared to the findings in the book?”_
> The book uses the same corpus to address whether CS brings about grammatical change, but it does not include CS metrics based in NLP research. This is the main and novel contribution of this paper: putting forward NLP CS metrics for the IU token level. Our experiments are the first to quantitatively validate the IU for CS metrics (M-Index, I-Index).
>
> - _“Assuming that your finding about the relevance of prosodic units is new, what was the reason for annotating them in the original corpus?”_
> IU-based transcription has been applied to other speech corpora, for example, the publicly available Santa Barbara Corpus of Spoken American English. IU-based transcription in the present corpus was used for grammatical analysis (for example, unexpressed (null) subjects in English are restricted to IU-initial position, as in:  And then I worked for a rancher over there for a while, ... Ø followed the rodeos for a while, (SBCSAE 32: 1587–1588) (Torres Cacoullos and Travis, 2018: 119-121)).
>
> - _“Could it be that the preference for code-switching at prosodic boundaries is simply caused by the Equivalence Constraint (because word order compatibility is highest at the boundary between two prosodic units)?”_
> This is a great question! The prosodic constraint is related to syntax in that words within prosodic boundaries have a tighter syntactic connection than words at prosodic boundaries. Therefore, the looser syntactic connection at the boundary between two prosodic units may mean greater word order flexibility. However, both within prosodic units and at prosodic unit boundaries, CS is subject to the Equivalence Constraint. Additionally,  among Equivalence points, CS is more likely at prosodic unit boundaries than within them. The prosodic constraint is thus independent of Equivalence.
>
> Lastly, we want to thank the reviewer for the stylistic suggestions as well as catching typos, which have been altered. We will refer to the CS visualizations as 'language distribution graphs' instead of heatmaps. We hope this distinguishes them from the usual heatmaps.

---

### Official Review · Reviewer_CGqL · 2023-08-07

**Typos Grammar Style And Presentation Improvements:** line 012
**Soundness:** 4

**Excitement:**

4: Strong: This paper deepens the understanding of some phenomenon or lowers the barriers to an existing research direction.

**Missing References:**

No

**Paper Topic And Main Contributions:**

The paper focuses on Code-Switching.
The authors argue that CS is driven by Intonation Units (and not by syntags), and that CS works differently between IU and inside IU.
They illustrate their claim by measuring the CS rate for 5 speakers from a Spanish-English database. This measure is based on the M-Index (0=one language, 1=balance of two languages) and the I-Index (rate of CS). For I-Index they adapt it to measure CS within IU and across IU.
The results show that "lone CS" (only one word is replace by another) is used within IU and "multiword CS" is used from a IU to another one.
Therefore, CS modeling should take into account these both behaviors.

**Questions For The Authors:**

CGqL_A. line 329-330 : I do not understand this sentence. Should I understand that is you have S and E in a IU, you keep only the first occuring, as you precise later?

CGqL_B. How could your work  be applied in a NLP task?

CGqL_C. The corpus is labeled with IU. Is it your work, or was initially the corpus labeled?

**Reasons To Accept:**

Paper should be accepted because:
- authors take time to explain the problem
- after reading one understands better that CS is not only a random modification of texts.
- their argue is very clear and easy to understand
- They describe M-Index and I-Index. This allows the paper to be self-contained
- the experiment illustrates well the argue.

**Reasons To Reject:**

I do not see reason to reject the paper.

**Reproducibility:**

4: Could mostly reproduce the results, but there may be some variation because of sample variance or minor variations in their interpretation of the protocol or method.

**Reviewer Confidence:**

4: Quite sure. I tried to check the important points carefully. It's unlikely, though conceivable, that I missed something that should affect my ratings.

---

> ### Author Rebuttal · Authors · 2023-08-27
>
> We thank Reviewer CGqL for these comments! We are glad to hear that they have no objections to accepting this paper into the conference, and find it easy to follow and exciting in its results. We are hopeful that we can clarify the questions below.
>
> The reviewer states towards the end of their summary that lone-items are used within an Intonation Unit (IU) and multi-word code-switching (CS) is used from one IU to another. Their formulation has inspired us to add the following sentences after line 458 to underscore this paper’s innovative results: “We put forward use of  the Intonation Unit as the token level for CS metrics. In addition, we calculate these metrics by distinguishing across-IU from within-IU CS, and multi-word CS from single-word (lone) other-language items. Application of the IU-based metrics to the data captures the tendency for multi-word CS to be used from one IU to another in across-IU CS, that is, the IU-Boundary Constraint of bilingual speech. In contrast, lone items by definition are used within a single IU and therefore raise the value of the I-Index for within-IU CS. Merging multi-word CS and lone items thus obscures the uniformity of the IU-Boundary Constraint among code switchers, despite individual differences in language distributions (M-Index) and CS probability (I-Index).”
>
> We are glad that CGqL thinks our experiment illustrates our argument well! However, we disagree that our work is a limited experiment. As noted in line 292, our work has a total of 41,000 words, and over 14,000 IU instances. See the table below for the breakdown of IU counts per transcript.
>
> | Transcript   | Number of IUs |
> | -------- | ------- |
> | 05  | 3,030    |
> | 27 | 3,181     |
> | 03    | 3,138    |
> | 16 | 2,639 |
> | 10 | 2,147 |
> | **TOTAL** | **14,135** |
>
> Our main constraint is that we chose monological transcripts to focus on the speaker, and not on how they respond to their interviewer. This arguably makes our findings more robust for our given domain (an individual's speech patterns), rather than simply ignoring the possible role of interlocutors in CS. Further work could investigate how CS metrics are impacted by dialogue, as suggested by some prior psycholinguistic and corpus-based work (e.g. by Kootstra and colleagues), although this is beyond the scope of this paper. We will be sure to further clarify the reasons for this constraint, which can be found in the original submission on lines 291-296.
>
> Next, we’d like to explicitly address each of the questions below.
>
> 1. _“line 329-330 : I do not understand this sentence. Should I understand that is you have S and E in a IU, you keep only the first occuring, as you precise later?”_
> These lines are meant to emphasize that although an IU can have multiple language tags (e.g., there's a switch within the IU), we do not want to double count an IU for any of the metrics. Therefore, lines 329-330 summarize our intent to transform the metrics into binary measures for the IU token. As clarified in lines 347-351, the M-Index rules are as follows: "When an IU has multiple languages contained within its bound, we tag it with the earliest language present (i.e., an IU with ‘SES’ will count as ‘S’). For the M-Index, we only considered ‘S’ and ‘E’ as valid language tags." In other words, we consider the earliest tag for the language representation for that IU. Lines 364-366 mention the measure for the Across-IU I-Index, and lines 374-375 have the rule for the Within-IU I-Index. We will work on rewriting lines 329-330 to clarify the above rules.
>
> 2. _"CGqL_B. How could your work be applied in a NLP task?"_
> This is a great question, which was also asked by Reviewer VWCM. Recall from lines 464-469: our main findings for CS metrics are that 1) the IU is a vital token level for switching metrics, and 2) there should be a distinction between single- and multi-word switching. We see relevance for this work in Controlled Natural Language Generation, specifically with the injection of linguistically well-founded (Reviewer VWCM) CS patterns to improve said generation (refer to lines 471-473). Currently, CS generation has focused on word substitutions (Solorio & Liu, 2008; Chang et al., 2019; Tarunest et al., 2021).  Importantly, our work is not limited to word substitution. In accounting for multi-word language alternations, we draw on the Equivalence Constraint (syntax) and the related but independent IU-Boundary Constraint (prosody). This paper is the first to use IUs for CS metrics.
> Ultimately, our work impacts the future construction of transcribed CS datasets. Much of the CS work is slow/iterative because it is hard to find quality CS data (refer to lines 195-197, where we cited (Doğruöz et al., 2021)). Our results, then, provide critical insight for future transcription and synthetic data generation methods that can improve code-switching datasets, which will ultimately impact all downstream NLP tasks.
>
> 3. _"CGqL_C. The corpus is labeled with IU. Is it your work, or was initially the corpus labeled?"_
> The corpus was initially labeled prior to our work as described by the corpus compilers in Torres Cacoullos and Travis (2018:49-51). We will be sure to clarify that prosodically-based transcription is part of the constitution of this corpus.
>
>
> Again, we’d like to thank Reviewer CGqL for the insightful comments, including the typo found in line 12 which we have fixed. Importantly, we will make sure to address our work’s applicability to NLP throughout the paper, including the literature review and conclusion. We thank the reviewer for their time spent on reading our work!

---

### Meta-Review · Area_Chair_RC8L · 2023-09-19

**Recommendation:** 5

**Metareview:**

The reviewers appreciated that the paper applies empirical data analysis methods to investigate linguistic phenomena, resulting better insights into the underlying linguistic phenomena (which could in turn be used to improve modeling of code-switching in data). The reviewers also note that the paper does an excellent job of introducing all of the relevant concepts, making the paper both clear and self-contained, even for a general NLP audience.

---

### Decision · Program_Chairs · 2023-10-07

**Decision:**

Accept-Main

**Comment:**

The reviewers appreciated that the paper applies empirical data analysis methods to investigate linguistic phenomena, resulting better insights into the underlying linguistic phenomena (which could in turn be used to improve modeling of code-switching in data). The reviewers also note that the paper does an excellent job of introducing all of the relevant concepts, making the paper both clear and self-contained, even for a general NLP audience.